# Understanding Rehabilitation Providers: Knowledge, Attitudes, and Practices Toward Older Adults with Substance Use Disorders

**Marybeth Johnson [1], Michelle L. Cathorall [2], Tina M. K. Newsham [3],\* and Elizabeth Fugate-Whitlock [4]**

1   Osher Lifelong Learning Institute, University of North Carolina Wilmington, Wilmington, NC 28403, USA; johnsonme@uncw.edu
2   School of Health and Applied Human Sciences, University of North Carolina Wilmington, Veterans Hall 2506I, Wilmington, NC 28403, USA; cathorallm@uncw.edu
3   School of Health and Applied Human Sciences, University of North Carolina Wilmington, McNeill Hall 2032, Wilmington, NC 28403, USA
4   School of Health and Applied Human Sciences, University of North Carolina Wilmington, McNeill Hall 2027, Wilmington, NC 28403, USA; whitlocke@uncw.edu
*   Correspondence: newshamt@uncw.edu

**Abstract**

**Objective**: The purpose of this study was to investigate the knowledge, attitudes, and practices (KAPs), including ageism, of rehabilitation service providers regarding older adults with substance use disorders to examine the association between KAPs and ageism on the knowledge of rehabilitation providers and confidence in treating this population. **Methods**: An online survey was developed to assess providers' familiarity with geriatric substance use disorders, attitudes towards aging, and perceived barriers to treatment. The survey included the Expectations Regarding Aging (ERA-12) tool to measure ageist attitudes. Data was collected from 25 rehabilitation healthcare providers across rehabilitation centers in North Carolina. Descriptive statistics and ERA-12 scoring were used to analyze the results. **Results**: Most (52.0%) respondents reported slight or moderate familiarity with specific risk factors for substance use disorders associated with older adults, and participants most commonly expressed ambivalence (48.0% indicated they were neither satisfied nor dissatisfied) with their training on this demographic. Barriers included a lack of specialized training, limited availability of age-appropriate treatment programs, and resistance to change. Negative attitudes towards aging and substance use disorders were prevalent among respondents. Providers indicated a need for enhanced education, clinical guidelines, and access to geriatric-trained professionals. **Discussion**: The findings highlight a critical need for specialized training for rehabilitation providers to improve care for older adults with substance use disorders. Addressing ageism, increasing awareness, and enhancing provider education are essential to improving treatment outcomes. Implementing targeted training programs and specialized resources could significantly enhance the quality of care for this underserved population.

**Keywords:** ageism; rehabilitation; healthcare providers; training; treatment barriers

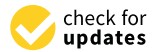

# 1. Introduction

A substance use disorder (SUD) is characterized as a mental condition that impacts an individual's cognitive functions and behavior, resulting in an incapacity to control the consumption of substances, including legal or illicit drugs, alcohol, and medications [1]. In 2022, there were 46.8 million Americans who met the DSM-5 criteria for having a

substance use disorder [2]. The DSM-5 diagnostic criteria are classified by the following four criterion types: impaired control over substance use, social impairment, risky use, and pharmacologic use [3]. Nationally, there was a 43% increase in drug-related fatalities between the periods of 2016–2018 and 2019–2021, and opioid-related fatalities played a significant role in driving this escalation in mortality rates [4].

An estimated 3.9 million adults aged 65 and older have a SUD, with 2.3 million diagnosed with an alcohol use disorder and about 1.8 million diagnosed with a drug use disorder in 2022 [2]. However, current statistics are an underestimation due to the lack of SUD assessments in older adults and the lack of self-report data by older adults.

## 1.1. Overview of Substance Use

Older adults are an often-overlooked demographic in the context of substance use disorders for a variety of reasons. With advancing age, the human body takes longer to metabolize alcohol and medications, extending the active presence of such substances in the system [5]. Up to 15% of health issues experienced by older adults can be attributed to the misuse of alcohol or prescription medications. Similarly, Arndt and colleagues [6] found that first-time admissions for individuals aged 55 and older have increased over time compared to their younger counterparts aged 30 to 54. This pattern was also observed in terms of admissions for older adults struggling with illicit drug-related issues. Zolopa and colleagues [7] discovered a similar trend with individuals in older cohorts initiating illicit drug use at higher rates than previous generations in community samples and rehabilitation treatment services, which has negative implications for harm reduction and rehabilitation treatment services. Furthermore, age-related health implications that drug use has on the physical wellbeing of older adults include inflammation, cellular aging, declining cognitive function, and reduced brain volume [7].

## 1.2. Barriers to Treatment

Treatments for substance abuse include behavioral interventions, residential rehabilitation, outpatient rehabilitation, and therapy programs. According to the U.S. Department of Health and Human Services [8], approximately 94% of people abusing substances in 2022 did not receive the treatment they needed. Demographics such as age and gender can influence the mode of diagnosis and treatment needed for an individual [9]. Stigma among health professionals towards patients with substance use disorders also has negative consequences for healthcare delivery, as health professionals have been found to lack adequate training and education in addition to having negative perceptions about patients with substance use disorders [10].

Ageism, stereotypes, prejudice, and discrimination directed toward a person based on their age [11] can influence how medicine is practiced by practitioners and can result in negative, but avoidable, health outcomes in older people [12]. Older adults often go undiagnosed and untreated due to a variety of factors, including stigma, lack of health screenings, resistance to treatment, misdiagnosis, limited access to appropriate care, age discrimination, and more. The number of untreated older adults with substance use disorders is underreported, exacerbated by these barriers [9]. Age discrimination was observed in the rehab centers by Wadd and Dutton [13], highlighting another barrier older adults could face when seeking treatment.

Another barrier to the treatment of older adults with SUD is the lack of awareness of when to seek treatment [14] and resistance to treatment [13]. Participants examined by Foster and colleagues revealed factors that contributed to the problem of being identified and seeking treatment as being related to health problems, social and financial stability, and legal issues [14].

### 1.3. Substance Use and Older Adults

Odani and colleagues [15] observed the misuse of prescription medications in adults aged fifty and older, noting that physicians often prescribe medications with addictive side effects for pain management, and that for older adults, the most common misused prescription drugs included sedatives, pain relievers, stimulants, and tranquilizers. This issue often goes unnoticed because people do not recognize that older adults can face the same substance abuse issues as younger individuals [8]. Prescription medication misuse can also be easier to conceal and harder to acknowledge in older adults due to misconceptions about substance use in this population [15].

Diagnosis and treatment of older people can be unique and require different strategies from those of younger people due to the aging process [16]. The DSM-5 criteria for the diagnosis of a substance use disorder [3] may be less relevant to older adults due to biological and social factors that may accompany advancing age. Physicians can mistake criteria for diagnosis as normal psychological and biological changes that happen as a person ages [16].

While it is crucial that mental health professionals are knowledgeable and skilled in this area to avoid misdiagnosis and mistreatment, Morgan et al. [17] highlighted that many healthcare professionals in this field are unprepared to address the needs of older adults. There is a need for specialized treatments and improved screening tools, given the increase in older adults with substance use disorders [18]. When providing interventions, rehabilitation providers need to be well educated in the specialized needs and care for older adults facing substance use problems and have the proper knowledge needed to provide treatment. The growing demand from this population will require new training adapted to the needs of older adults seeking treatment [19].

### 1.4. Outcomes of Treatment for Older Adults

There is limited research on the follow-up treatment of older adults with substance use disorders. Dingle and Bowen [20] found that older adults had higher rates of abstinence and success in these relapse prevention programs compared to the younger age group. Similarly, physician/provider counseling for older adults helped reduce at-risk drinking behaviors and increase the success rates of interventions and can be beneficial for the overall health outcomes of older adults needing interventions [19]. To carry out these interventions, providers would need to be well educated in the specialized needs and care for older adults facing substance use problems and have the proper tools needed to provide treatment [19].

There is a need to bridge the knowledge gap in the treatment of older adults, particularly among healthcare professionals operating in rehabilitation facilities where individuals are seeking assistance and care. The purpose of the current study was to examine the knowledge, attitudes, and practices (KAPs), including ageism, of rehabilitation service providers regarding older adults with substance use disorders to examine the association of KAPs and ageism on providers knowledge and confidence treating this population The findings from this research add to the body of knowledge related to improving the quality of care received by older adults from rehabilitation services providers.

## 2. Materials and Methods

### 2.1. Participants

A cross-sectional study design, using a combination of convenience and snowball sampling techniques to recruit participants, was used. Facilities in North Carolina that met the inclusion criteria (providing rehabilitation treatment services for substance use and offering an online public contact form) were contacted. Recruitment began with the distribution of emails to thirty-six rehabilitation/treatment centers across North Carolina based on a Google search. The initial email provided a summary of the study along with a direct hyperlink to the survey,

which was distributed through the Qualtrics survey platform. The employee who received the initial email that was sent through each site's general contact form initiated snowball sampling by forwarding the message to the employee listserv or spreading the word about participating in the study throughout their place of employment. To improve response rates after the initial email, follow-up emails were sent a week later. The survey remained active for the three-week recruitment period. When the researchers received employee responses inquiring about the distribution of the survey, the researchers requested that the email/survey link be sent out to all rehabilitation employees. Consent was gathered through agreement with a consent statement at the beginning of the survey.

### 2.2. Measures

A survey was developed to gather data from participants. Part one of the survey included ten demographic questions: occupational title, highest level of training/certification, and years of experience working with older adults.

### 2.2.1. Knowledge, Attitudes, and Practices

There is currently no validated tool for measuring rehabilitation providers' knowledge, attitudes, and practices in their approach to working with older adults with substance use disorders. As a result, an instrument was developed for this study based on key factors identified in the literature that influence providers' work with older adults experiencing substance use disorders. The tool comprised eighteen questions, including seventeen Likert scale items and one ranking order question (see Appendix A). Likert items asked participants to rate their confidence in their ability when treating older adults with substance use disorders, as well as their confidence in their knowledge and attitudes related to working with older adults. To assess provider perceptions of what they believed to be the challenges older adults with substance use disorders face in accessing and engaging with rehabilitation services, participants were asked to select all that applied in a list of choices. The survey was vetted by a committee of three faculty members in gerontology and public health, but it was not a validated instrument.

### 2.2.2. Ageism

To assess ageism among respondents, the Expectations Regarding Aging (ERA) was utilized [21]. The 12-Item ERA is a validated tool to measure participant expectations regarding aging in the physical health, mental health, and cognitive function domains. The ERA has adequate content validity, structural validity, and internal consistency (Cronbach's coefficient alpha $\leq$ 0.73) [21]. There were no cutoff scores; for each domain in the ERA and the overall global score, the range was 0–100. Higher scores indicate higher expectations regarding aging (i.e., more positive beliefs about growing older—lower ageism) in each domain, and lower scores indicate lower expectations regarding aging (i.e., more negative beliefs about growing older—higher ageism) in each domain.

### 2.3. Procedures

Participants completed this cross-sectional survey upon receipt of the invitation email. Because all data were collected anonymously, no contact information was gathered, nor was any follow-up conducted with participants upon completion of the survey, and the risks to participants were no more significant than faced in everyday life, this study was determined to be exempt from the University of North Carolina Wilmington Institutional Review Board (IRB# H24-0434).

*2.4. Analysis*

Data was analyzed using Excel. Descriptive statistics were run for all questions in the survey. Descriptive statistics are presented as frequencies and percentages in the tables. The ERA-12 was scored according to the authors' instructions for each of the three domains: physical health, mental health, and cognitive function, described above in Section 2.2. The overall scores of each participant were also calculated per the scoring instructions published with the instrument [21]. Spearman's Rank Order Correlations between the ERA-12 and questions related to participants' knowledge, attitudes, and practices were conducted to explore the association between healthcare providers' expectations regarding aging and the medical care and services they provide to older adults. Spearman's correlation was the most appropriate for the nonparametric, ordinal-level data. There were no statistically significant relationships identified.

## 3. Results

The sample included 25 participants employed at rehabilitation/treatment centers in North Carolina. Out of the 25 participants, one did not complete part three of the survey, which included the Expectations Regarding Aging (ERA) questions, resulting in a sample of 24 participants for the analysis of the ERA. Most respondents were Caucasian (*n* = 20; 80%), female (*n* = 20; 80%), and employed full-time (*n* = 22; 88%). The average age of the sample was 40.9 years, and most participants had earned a master's degree (*n* = 20; 80%). The sample represented a diverse range of occupations in a rehabilitation setting. See Table 1 for more demographic details.

*3.1. Knowledge, Attitudes, and Behaviors*

Although almost half (*n* = 12; 48%) of the participants reported that they often or very often engaged with older adults with SUD in their work (see Table 2), over half (*n* = 13; 52%) indicated that they rarely or occasionally came into contact with older adults with substance use disorders in practice. Participants reported higher confidence levels in treating adults compared to adolescents (age 12–17) or older adults (age 65 and older). For instance, 72.0% (*n* = 18) of respondents strongly agreed with feeling confident in treating adults (age 18–64), while only 40.0% (*n* = 10) expressed the same level of confidence in treating older adults. There were also disparities in confidence levels in providers' knowledge of older adults and the specialized needs of the age group compared to younger adults (i.e., adults under age 65). While over half of the participants, 56.0% (*n* = 14), agreed that they understood the specialized needs of older adults, 44.0% (*n* = 11) reported that they were less confident with their knowledge of older adults. For more details, see Table 3.

Regarding barriers to treating older adults with SUD, the most cited challenges were the stigma associated with aging and substance abuse (*n* = 17; 21.8%), with lack of awareness about available services (*n* = 16; 20.5%) and physical health limitations (*n* = 15; 19.2%) also being identified frequently as barriers. For more information, see Table 4.

Regarding participants' perception of how best to treat substance abuse among older adults, individual counseling was reported to be the most effective strategy/intervention for older adults with substance use disorders, with 42.9% (*n* = 9) ranking this first out of five choices. Group therapy was also indicated to be effective, with 23.8% (*n* = 5) ranking as first and 33.3% (*n* = 7) as second options. For more information, see Table 5.

**Table 1.** Demographic characteristics of participants.

| Variable | *n* = 25 |
|---|---|
| | *M* **(Range)** |
| **Age of participants** | 40.9 (24–74) |
| | |
| **Race** | *n* **(%)** |
| *Caucasian* | 20 (80.0%) |
| *African American* | 3 (12.0%) |
| *Latino or Hispanic* | 1 (4.0%) |
| *Two or more* | 1 (4.0%) |
| **Gender** | |
| *Female* | 20 (80.0%) |
| *Male* | 5 (20.0%) |
| **Education** | |
| *Bachelor's Degree* | 3 (12.0%) |
| *Master's Degree* | 20 (80.0%) |
| *Doctorate or Higher* | 2 (8.0%) |
| **Employment** | |
| *Employed, working 1–39 h per week* | 3 (12.0%) |
| *Employed, working 40 or more hours per week* | 22 (88.0%) |
| **Years of experience** | |
| *In reported occupation* | 13.9 (1–40) |
| *Working with older adults* | 13.0 (0–40) |
| **Job Title** | |
| *Social worker* | 6 (30%) |
| *Licensed therapist/counselor* | 5 (25%) |
| *Registered nurse* | 4 (20%) |
| *Program supervisor* | 4 (20%) |
| *Clinical director* | 3 (15%) |
| *Physician* | 2 (10%) |
| *Care manager* | 1 (5%) |

Note: Bold indicates variables and italics indicates response options.

**Table 2.** Frequency of contact with older adults with substance use disorders in rehabilitation practices.

| Item | *n* (%) Indicating Frequency of Encounters |
|---|---|
| Rarely | 4 (16.0%) |
| Occasionally | 9 (36.0%) |
| Often | 10 (40.0%) |
| Very often | 2 (8.0%) |

**Table 3.** Participant's knowledge, attitudes, and practices when treating older adults with substance use disorders.

| Item / Response | Strongly Agree *n* (%) | Agree *n* (%) | Neither Agree nor Disagree *n* (%) | Disagree *n* (%) | Strongly Disagree *n* (%) |
|---|---|---|---|---|---|
| I feel confident in my ability to treat adolescents. | 7 (28.0%) | 6 (24.0%) | 3 (12.0%) | 9 (36.0%) | 0 (0.0%) |
| I feel confident in my ability to treat adults. | 18 (72.0%) | 6 (24.0%) | 1 (4.0%) | 0 (0.0%) | 0 (0.0%) |
| I feel confident in my ability to treat older adults. | 10 (40.0%) | 12 (48.0%) | 1 (4.0%) | 2 (8.0%) | 0 (0.0%) |
| I feel confident in my knowledge of older adults. | 9 (36.0%) | 11 (44.0%) | 3 (12.0%) | 2 (8.0%) | 0 (0.0%) |
| I understand older adults have specialized needs compared to younger adults. | 9 (36.0%) | 14 (56.0%) | 2 (8.0%) | 0 (0.0%) | 0 (0.0%) |
| I have difficulty understanding the needs of older adults. | 0 (0.0%) | 2 (8.0%) | 4 (16.0%) | 14 (56.0%) | 5 (20.0%) |

**Table 4.** Perception regarding challenges older adults with substance use disorders face in accessing and engaging with rehabilitation services.

| Item | n (%) Indicating This Item Is a Challenge |
|---|---|
| Stigma associated with aging and substance abuse | 17 (21.8%) |
| Lack of awareness about available services | 16 (20.5%) |
| Physical health limitations | 15 (19.2%) |
| Transportation difficulties | 11 (14.1%) |
| Mental health comorbidities | 10 (12.8%) |
| Financial constraints | 8 (10.3%) |
| Other | 1 (1.3%) |

**Table 5.** Perceived effectiveness of strategies and interventions to treat older adults with substance use disorders.

| Strategy | n (%) Rank 1 | n (%) Rank 2 | n (%) Rank 3 | n (%) Rank 4 | n (%) Rank 5 | n (%) Rank 6 |
|---|---|---|---|---|---|---|
| Individual counseling | 9 (42.9%) | 7 (33.3%) | 2 (9.5%) | 2 (9.5%) | 1 (4.8%) | 0 (0.0%) |
| Group therapy | 5 (23.8%) | 7 (33.3%) | 4 (19.0%) | 3 (14.3%) | 2 (9.5%) | 0 (0.0%) |
| Medication-assisted treatment | 2 (9.5%) | 6 (28.6%) | 5 (23.8%) | 4 (19.0%) | 4 (19.0%) | 0 (0.0%) |
| Harm reduction approaches | 4 (19.0%) | 0 (0.0%) | 3 (14.3%) | 9 (42.9%) | 5 (23.8%) | 0 (0.0%) |
| Family therapy | 1 (4.8%) | 1 (4.8%) | 7 (33.3%) | 3 (14.3%) | 9 (42.9%) | 0 (0.0%) |
| Other | 0 (0.0%) | 0 (0.0%) | 0 (0.0%) | 0 (0.0%) | 0 (0.0%) | 21 (100.0%) |

Over half of the participants (*n* = 13; 52.0%) were only slightly or moderately familiar with specific risk factors associated with substance use disorders in the older adult population (Table 6). A minority (*n* = 10; 40.0%) of respondents were either somewhat satisfied or extremely satisfied with their training/credential process regarding education on older adults. However, most respondents were neither satisfied nor dissatisfied with the training or credentialing process regarding education of older adult patients (*n* = 12; 48.0%; Table 7). For treatment of patients with substance use disorders, 64.0% (*n* = 16) selected that they slightly modify their approach when working with older adults compared to younger individuals (Table 8). The highest-ranked barrier perceived in effectively addressing the needs of older adults was a lack of specialized training in geriatric substance use disorders. Limited availability of age-appropriate treatment programs (*n* = 12; 24.0%) and resistance to change or treatment adherence issues (*n* = 11; 22.0%) were also ranked as high barriers (Table 9).

**Table 6.** Participant familiarity with specific challenges and risk factors associated with substance use disorders in the older adult population.

| Item | n (%) Indicating Familiarity with Challenges |
|---|---|
| Not familiar at all | 0 (0.0%) |
| Slightly familiar | 5 (20.0%) |
| Moderately familiar | 8 (32.0%) |
| Very familiar | 11 (44.0%) |
| Extremely familiar | 1 (4.0%) |

**Table 7.** Satisfaction rating with the training or credentialing process regarding education of older adult patients.

| Item | *n* (%) Indicating Satisfaction with Training |
|---|---|
| Extremely dissatisfied | 0 (0.0%) |
| Somewhat dissatisfied | 3 (12.0%) |
| Neither satisfied nor dissatisfied | 12 (48.0%) |
| Somewhat satisfied | 8 (32.0%) |
| Extremely satisfied | 2 (8.0%) |

**Table 8.** Adaptation of approaches when working with older adults compared to younger individuals with substance use disorders.

| Item | *n* (%) Indicating Adapting Approaches |
|---|---|
| I use the same approach for both age groups | 0 (0.0%) |
| I modify my approach slightly for older adults | 16 (64.0%) |
| I significantly modify my approach for older adults | 7 (28.0%) |
| I use a completely different approach for older adults | 2 (8.0%) |

**Table 9.** Perceived barriers in effectively addressing the unique needs of older adults with substance use disorders.

| Item | *n* (%) Indicating This Item Is a Barrier |
|---|---|
| Lack of specialized training in geriatric substance use disorders | 17 (34.0%) |
| Limited availability of age-appropriate treatment programs | 12 (24.0%) |
| Resistance to change or treatment adherence issues | 11 (22.0%) |
| Ageism within the healthcare system | 7 (14.0%) |
| Other | 3 (6.0%) |

Respondents reported that continuing education workshops/seminars (*n* = 17; 30.9%), clinical guidelines specific to older adults with substance use disorders (*n* = 10; 20.0%), and access to specialized geriatrics-trained addiction counselors/providers (*n* = 10; 20.0%) would best enhance their ability to support older adults in rehabilitation (Table 10). When asked how attitudes towards substance use disorders in older adults impact the effectiveness of rehabilitation efforts, participants most commonly (*n* = 12;48.0%) reported negative impacts. There was also a significant percentage of participants who reported that they were unsure of the effects these attitudes had on the effectiveness of rehabilitation efforts for older adults (*n* = 8; 32.0%; Table 11).

**Table 10.** Resources or training participants believe would enhance their ability to support older adults with substance use disorders in rehabilitation.

| Item | *n* (%) Indicating Item Would Be Beneficial |
|---|---|
| Continuing education workshops/seminars | 17 (30.9%) |
| Online courses/modules | 10 (18.2%) |
| Mentorship programs | 4 (7.3%) |
| Clinical guidelines specific to older adults with substance use disorders | 11 (20.0%) |

**Table 10.** *Cont.*

| Item | n (%) Indicating Item Would Be Beneficial |
|---|---|
| Access to specialized geriatric addiction counselors | 11 (20.0%) |
| Other | 2 (3.6%) |

**Table 11.** Participants believe that attitudes towards substance use disorders in older adults impact the effectiveness of rehabilitation efforts.

| Item | n (%) Indicating Effects of Attitudes |
|---|---|
| Positively | 5 (20.0%) |
| Negatively | 12 (48.0%) |
| No impact | 0 (0.0%) |
| Unsure | 8 (32.0%) |

*3.2. Expectations Regarding Aging*

Findings of the ERA-12 survey revealed that the majority of participants ($n = 17$; 66.7%) had a mean score of 61.5% ($SD = 18.9$) on the overall scale. For the physical health scale, 54.2% ($n = 14$) of the sample scored at or below 50 on the scoring scale. Similarly, in the cognitive function scale, 41.6% ($n = 10$) scored at or below 50, and 25.0% ($n = 6$) scored at or below 50 in the mental health domain. These findings, captured in Table 12, indicate higher expectations of older adults' mental health compared to participants' expectations of older adults' physical and cognitive functioning.

**Table 12.** Mean Expectations Regarding Aging Scale Results.

| Scale | M (SD) |
|---|---|
| Physical health | 52.8 (18.7) |
| Mental health | 72.2 (22.0) |
| Cognitive function | 60.0 (25.8) |
| Overall (Combined domain scores) | 61.5 (18.9) |

## 4. Discussion

The researchers in this study examined the knowledge, attitudes, and practices of rehabilitation professionals when engaging with older adults with substance use disorder to explore providers' preparation to adequately treat the specialized needs of older adults. While previous researchers examined ageism and older adult care among nursing, social work, and physician healthcare professionals, this work represents a novel attempt to gather data from participants from seven different occupational titles within rehabilitation services. Ben-Harush and colleagues [22] found themes of inadequate treatment of older adult patients in the healthcare setting by health professionals due to patients' age. Unsurprisingly, participants in the current study held lower expectations regarding older adults' physical and cognitive functioning, as ERA-12 scoring states that lower scores on the 0–100 scale indicate lower overall expectations regarding aging [21]. The presence of higher expectations regarding older adults' mental health may indicate that rehabilitation providers may erroneously hold ageist beliefs that impact their treatment of older adult patients/clients by assuming that older people are less susceptible to mental health challenges such as substance use disorders.

Another notable finding is the variance in confidence levels reported by participants in treating different age groups. While participants reported a high level of confidence in treating adults, their confidence diminished when it came to treating older adults. For

example, 72.0% of respondents strongly agreed with feeling confident in treating adults, while only 40.0% expressed the same level of confidence in treating older adults. This discrepancy suggests a potential area for targeted training and professional development initiatives to bolster confidence in addressing the unique needs of older adults with substance use disorders. Training healthcare providers with specific knowledge and skills has been effective at increasing confidence in communicating about end-of-life care plans [23], patient-centered communication [24], and delivering quality care [25]. The challenges identified by providers in accessing and engaging with rehabilitation services for older adults with substance use disorders underscore the multifaceted nature of barriers faced by this population. Stigma, lack of awareness about available services, and physical health limitations emerged as primary barriers, highlighting the importance of addressing clinical needs, societal attitudes, and systemic gaps in service provision.

Perceived barriers to addressing the needs of older adults included a lack of specialized training, limited availability of age-appropriate treatment programs, and resistance to change or treatment adherence issues. These barriers underscore the importance of systemic changes in healthcare delivery, including expanded training opportunities, increased access to specialized care, and the development of clinical guidelines tailored to older adults with substance use disorders. This study also revealed notable knowledge gaps among participants, particularly regarding specific risk factors associated with substance use disorders in older adults. Despite a moderate level of satisfaction with training and credentialing processes, there remains a need for enhanced education and resources focused on geriatric substance use disorders to bridge these gaps effectively. Professional development training could mitigate these barriers by providing the knowledge and skills necessary for effective communication and clinical practice specific to older adults with substance use disorders.

Attitudes towards substance use disorders in older adults were predominantly negative among participants, highlighting the potential impact of stigma on rehabilitation efforts. Addressing these negative attitudes through education, advocacy, and destigmatization efforts is crucial to support the needs of older adults seeking treatment. The finding that over half (52.0%) of participants rarely or occasionally encountered older adults with substance use disorders in their practice raises questions about accessibility and outreach efforts targeting this population, which may stem from beliefs that older adults are less susceptible to such disorders. As rates of SUD and first-time admission to rehabilitation for illicit drug use are increasing among older people [6,7], rehabilitation providers will continue to see more older adults with unique presentations and challenges, and providers need to be prepared to deliver appropriate services. Providers' lack of knowledge of aging and lack of awareness of older adult substance misuse may lead providers to assume impairments they see are due to age and not substance misuse (i.e., providers expect to see physical decline in older adults as indicated by ERA-12 physical domain scores). The highest-ranked barriers (42.3%) emphasized the need for decreased stigma associated with aging and increased awareness about available services for older adults.

In addition to being familiar with barriers older adults face, results indicate that providers could use additional information and education on the risk factors associated with substance use disorders in the older population. This was supported by most respondents (52.0%) reporting being only slightly or moderately familiar with the specific risk factors associated with substance use disorders in the older adult population. Perceived barriers from participants in addressing the unique needs of older adults revealed that specialized training in geriatric substance use disorders is needed. Enhancing screening protocols, increasing awareness among older adults and their caregivers, and promoting age-appropriate interventions are essential steps towards addressing this gap in service

provision. The limited availability of age-appropriate treatment programs and resistance to change or treatment adherence issues were also indicated as barriers by respondents.

Participants' limited confidence in their ability to treat older people indicates a need for specialized training for rehabilitation service providers to prepare them to work with older adults with SUD with confidence. Targets for such education could include knowledge about aging and the impact it has on older adults' physical and cognitive abilities compared to adult populations, providing side-by-side case studies of adults and older adults to enable providers to differentiate how substance use intersects with aging to create different needs for older adults. The outcomes of the training could be increased confidence levels in providers' ability to treat older adults and increased confidence in their knowledge of older adults. Educating providers on the specialized needs of older adults with substance use disorders can help prevent or mitigate barriers to treatment for them. Participants indicated continuing education workshops/seminars (30.9%), clinical guidelines specific to older adults with substance use disorders (20.0%), and access to specialized geriatrics-trained addiction counselors/providers (20.0%) would enhance their ability to support older adults with substance use disorders in rehabilitation. This information can be provided to rehabilitation centers to motivate and support the implementation of educational workshops/seminars for healthcare employees encountering older adults.

## 5. Limitations and Suggestions

This study was among the first to investigate the knowledge, attitudes, and practices of rehabilitation services providers working with older adults with substance use disorders. Due to this, much of the survey used in this study was developed based on existing literature. Therefore, the questionnaire was not validated prior to use, limiting confidence in the findings of this study. Suggestions for future research endeavors would include adjusting the language of some of the survey questions for clarity and then testing the validity and reliability of the survey.

The small sample size is also a limitation. With a sample of only twenty-five people, the sample does not represent all rehabilitation service providers. Additional research is needed to use the survey tool developed in this study alongside other validated measures to better understand the knowledge, attitudes, and practices of different rehabilitation providers working with older adults who have substance use disorders. These assessments are important for identifying the challenges providers face and for improving the care and treatment that older adults with substance use disorders receive.

There was response bias in that the people who responded felt some motivation to participate in a study on SUD treatment of older adults. Due to this, their scores likely underrepresent the prevalence of ageism and lack of confidence in knowledge and treatment ability among providers. Further, response on self-report/self-administered surveys bias (also called decision style) of this type has been reported [26]. A future study could examine a larger randomly selected sample that would be representative of the service providers caring for older adults with SUD to gain a more accurate understanding of rehabilitation providers' knowledge, attitudes, and practices of treating older adults with SUD. With a larger sample, methods for controlling for response bias/decision style could also be applied [26]. The findings from a larger study could also be used to develop a training program for rehabilitation providers that could be tested in a future study to determine effectiveness at increasing knowledge and confidence levels of providers working with older adults.

## 6. Conclusions

There is a critical need for specialized training among rehabilitation service providers to effectively address the unique needs of older adults with substance use disorders.

Insights gained from this study hold the potential to enhance the quality of care provided by rehabilitation service providers to older adults. By enhancing providers' knowledge and confidence levels, such training initiatives can contribute to overcoming barriers to treatment and ensuring older adults receive the support this unique demographic requires. The importance of combating stigma associated with aging, increasing awareness about available services, and educating providers on the specific risk factors inherent in substance use disorders among the older adult population cannot be overstated. Furthermore, the identified preference for continuing education workshops, clinical guidelines, and access to specialized professionals highlights possible avenues for improving care delivery. Although there is a need for profession-specific training, the approaches of social workers differ from those of a therapist or registered nurse, and tools such as Ageism First Aid training from the Gerontological Society of America [27] can be utilized for the generalized training of health professionals. By integrating these insights into rehabilitation practices, researchers, educators, and policymakers can better equip providers to navigate the complexities of geriatric substance use disorders, ultimately enhancing the quality of care and outcomes for older adults in need.

**Author Contributions:** M.J., as a master's student, developed and implemented this research project under the guidance of her advisory committee (the co-authors). She developed and deployed the survey, gathered and analyzed the data, and wrote up the findings. M.L.C. served as a reader for M.J.'s master's project committee and contributed to the development of the methods and guided the literature review, data analysis, and write-up of the findings. T.M.K.N. served as the committee chair for M.J.'s master's project and contributed to the development of the methods and guided the literature review and write-up of the findings. E.F.-W. served as a committee reader for M.J.'s master's project and contributed to the development of the methods and guided the literature review and write-up of the findings. All authors have read and agreed to the published version of the manuscript.

**Funding:** This research received no external funding.

**Institutional Review Board Statement:** The University of North Carolina Wilmington Institutional Review Board (IRB) determined that this study was exempt from ethical approval (IRB# H24-0434).

**Informed Consent Statement:** Not applicable.

**Data Availability Statement:** Data are not available due to ethical restrictions. Please contact corresponding author with inquiries.

**Conflicts of Interest:** The authors report no conflicts of interest.

## Appendix A  Survey Tool

You are invited to complete an anonymous online survey that will take approximately 5–10 min. The survey does not collect any sensitive or personally identifiable information and is intended solely for research purposes. Participation is entirely voluntary, and you may choose to stop at any time without penalty.

Q1. How old are you?

___________________________________________________________

Q2. What county do you live in?

___________________________________________________________

Q3. With which racial/ethnic group do you identify?

- ○ Caucasian (1)
- ○ African American (2)
- ○ Latino or Hispanic (3)
- ○ Asian (4)
- ○ Native American (5)

- Native Hawaiian or Pacific Islander (6)
- Two or more (7)
- Other/Unknown (8)
- Prefer not to say (9)

Q4. With which gender do you identify?

- Male (1)
- Female (2)
- Other: (3)
- Prefer not to say (4)

Q5. What is the highest level of education you have completed?

- Some High School (1)
- High School (2)
- Some College (3)
- Associate's Degree (4)
- Bachelor's Degree (5)
- Master's Degree (6)
- Doctorate or higher (7)
- Trade School (8)
- Prefer not to answer (9)

Q6. What is your occupational title?

_______________________________________________

Q7. How many years of experience do you have in this occupation?

_______________________________________________

Q8. What is the highest level of training or certification you have received?

_______________________________________________

Q9. How many years of experience do you have working with adults aged fifty-five years and older?

_______________________________________________

Q10. What is your employment status?

- Employed, working 1–39 h per week (1)
- Employed, working 40 or more hours per week (2)

Q11. I feel confident in my ability to treat adolescents.

- Strongly agree (1)
- Agree (2)
- Neither agree nor disagree (3)
- Disagree (4)
- Strongly disagree (5)

Q12. I feel confident in my ability to treat adults.

- Strongly agree (1)
- Agree (2)
- Neither agree nor disagree (3)
- Disagree (4)
- Strongly disagree (5)

Q13. I feel confident in my ability to treat older adults.

- Strongly agree (1)
- Agree (2)
- Neither agree nor disagree (3)
- Disagree (4)

○ Strongly disagree (5)

Q14. I feel confident in my knowledge of older adults.

○ Strongly agree (1)
○ Agree (2)
○ Neither agree nor disagree (3)
○ Disagree (4)
○ Strongly disagree (5)

Q15. I understand older adults have specialized needs compared to younger adults.

○ Strongly agree (1)
○ Agree (2)
○ Neither agree nor disagree (3)
○ Disagree (4)
○ Strongly disagree (5)

Q16. I have completed training that expanded my knowledge of older adults.

○ Strongly agree (1)
○ Agree (2)
○ Neither agree nor disagree (3)
○ Disagree (4)
○ Strongly disagree (5)

Q17. I have difficulty understanding the needs of older adults.

○ Strongly agree (1)
○ Agree (2)
○ Neither agree nor disagree (3)
○ Disagree (4)
○ Strongly disagree (5)

Q18. How would you rate the training of healthcare professionals in identifying and addressing substance use disorders in older adults?

○ Satisfactory (1)
○ Somewhat satisfactory (2)
○ Neither satisfactory nor unsatisfactory (3)
○ Somewhat unsatisfactory (4)
○ Unsatisfactory (5)

Q19. How familiar are you with the specific challenges and risk factors associated with substance use disorders in the older adult population?

○ Not familiar at all (1)
○ Slightly familiar (2)
○ Moderately familiar (3)
○ Very familiar (4)
○ Extremely familiar (5)

Q20. How satisfied are you with your training or credentialing process regarding education of older adult patients?

○ Extremely dissatisfied (1)
○ Somewhat dissatisfied (2)
○ Neither satisfied nor dissatisfied (3)
○ Somewhat satisfied (4)
○ Extremely satisfied (5)

Q21. How confident are you in the awareness of healthcare providers regarding potential interactions between commonly prescribed medications for older adults and substances of abuse?

- ○   Very confident (1)
- ○   Confident (2)
- ○   Neutral (3)
- ○   Not confident (4)
- ○   Very unconfident (5)

Q22. How frequently do you encounter older adults with substance use disorders in your rehabilitation practice?

- ○   Rarely (1)
- ○   Occasionally (2)
- ○   Often (3)
- ○   Very often (4)

Q23. Which of the following challenges do you believe older adults with substance use disorders face in accessing and engaging with rehabilitation services?

- ○   Stigma associated with aging and substance use (1)
- ○   Lack of awareness about available services (2)
- ○   Physical health limitations (3)
- ○   Mental health comorbidities (4)
- ○   Financial constraints (5)
- ○   Transportation difficulties (6)
- ○   Other (please specify) (7)

Q24. Rate the effectiveness of the following strategies or interventions when working with older adults with substance use disorders: (1 being the highest, 6 being the lowest)

______ Individual counseling (1)
______ Group therapy (2)
______ Medication-assisted treatment (3)
______ Harm reduction approaches (4)
______ Family therapy (5)
______ Other (please specify) (6)

Q25. How do you adapt your approach when working with older adults compared to younger individuals with substance use disorders?

- ○   I use the same approach for both age groups (1)
- ○   I modify my approach slightly for older adults (2)
- ○   I significantly modify my approach for older adults (3)
- ○   I use a completely different approach for older adults (4)

Q26. What barriers, if any, do you perceive in effectively addressing the unique needs of older adults with substance use disorders?

- ○   Lack of specialized training in geriatric substance use disorders (1)
- ○   Limited availability of age-appropriate treatment programs (2)
- ○   Resistance to change or treatment adherence issues (3)
- ○   Ageism within the healthcare system (4)
- ○   Other (please specify) (5)

Q27. Which resources or training do you believe would enhance your ability to support older adults with substance use disorders in rehabilitation?

- ○   Continuing education workshops/seminars (1)
- ○   Online courses/modules (2)

- ○ Mentorship programs (3)
- ○ Clinical guidelines specific to older adults with substance use disorders (4)
- ○ Access to specialized geriatric addiction counselors (5)
- ○ Other (please specify) (6)

Q28. How do you believe attitudes towards substance use disorders in older adults impact the effectiveness of rehabilitation efforts?

- ○ Positively (1)
- ○ Negatively (2)
- ○ No impact (3)
- ○ Unsure (4)

Q29. When people get older, they need to lower their expectations of how healthy they can be.

- ○ Definitely true (1)
- ○ Somewhat true (2)
- ○ Somewhat false (3)
- ○ Definitely false (4)

Q30. The human body is like a car: When it gets old, it gets worn out.

- ○ Definitely true (1)
- ○ Somewhat true (2)
- ○ Somewhat false (3)
- ○ Definitely false (4)

Q31. Having more aches and pains is an accepted part of aging.

- ○ Definitely true (1)
- ○ Somewhat true (2)
- ○ Somewhat false (3)
- ○ Definitely false (4)

Q32. Every year that people age, their energy levels go down a little more.

- ○ Definitely true (1)
- ○ Somewhat true (2)
- ○ Somewhat false (3)
- ○ Definitely false (4)

Q33. I expect that as I get older, I will spend less time with friends and family.

- ○ Definitely true (1)
- ○ Somewhat true (2)
- ○ Somewhat false (3)
- ○ Definitely false (4)

Q34. Being lonely is just something that happens when people get old.

- ○ Definitely true (1)
- ○ Somewhat true (2)
- ○ Somewhat false (3)
- ○ Definitely false (4)

Q35. As people get older, they worry more.

- ○ Definitely true (1)
- ○ Somewhat true (2)
- ○ Somewhat false (3)
- ○ Definitely false (4)

Q36. It's normal to be depressed when you are old.

- ○ Definitely true (1)
- ○ Somewhat true (2)
- ○ Somewhat false (3)
- ○ Definitely false (4)

Q37. I expect that as I get older, I will become more forgetful.

- ○ Definitely true (1)
- ○ Somewhat true (2)
- ○ Somewhat false (3)
- ○ Definitely false (4)

Q38. It's an accepted part of aging to have trouble remembering names.

- ○ Definitely true (1)
- ○ Somewhat true (2)
- ○ Somewhat false (3)
- ○ Definitely false (4)

Q39. Forgetfulness is a natural occurrence just from growing old.

- ○ Definitely true (1)
- ○ Somewhat true (2)
- ○ Somewhat false (3)
- ○ Definitely false (4)

Q40. It is impossible to escape the mental slowness that happens with aging.

- ○ Definitely true (1)
- ○ Somewhat true (2)
- ○ Somewhat false (3)
- ○ Definitely false (4)

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
