# Peer review of "Understanding Rehabilitation Providers: Knowledge, Attitudes, and Practices Toward Older Adults with Substance Use Disorders"

_2673-9259, doi:10.3390/jal5040041_

Round 1

Reviewer 1 Report

Comments and Suggestions for Authors

Thank you for this study and paper! Your study hits on some very important points about rehabilitation providers’ knowledge, attitudes and practices towards older adults and particularly those with substance use disorders. There is no doubt a critical need to pay attention to your findings and suggestions.

Throughout the paper, there are points or sections needing clarification. First and foremost, the entire paper needs to be edited for sentence structure, grammar, and statistical references which I will mention later. Editing or sentence clarification needs to include the abstract (lines 26-28 – not clear if something is missing or if the sentence needs to be rewritten).

  • Although the journal has free style formatting for the presentation of content, it seems odd that sections 2 to 5 are not under the Introduction. The content for all of the sections 1 through 5 present the context and background to the paper very well but seem to be suited to be within the Introduction section as opposed to separate sections. Why are these independent sections?
  • Based on your introduction to older adults as being 65 and older (American Addictions Centers, 2023). Is this what you use as your guide throughout the paper when you refer to older adults? Need to clearly indicate if you are using this same guide or what other age range constitutes ‘older adults’ in your study
  • Within section 3 – Barriers to Treatment – mention was made in the opening lines (i.e. 77-79) that “…people abusing substances in 2022 did not receive the treatment they needed”. This seems to be aligned with the challenges most care services and patients experienced during COVID – was what you mentioned because of COVID? Please clarify the reason why people did not receive treatment needed.
  • As part of 6.1 Participants – you mentioned inclusion criteria for the facilities you contacted but no inclusion criteria for the participants. The latter would seem to be equally important - what criteria did you use to recruit and select participants? For example, in the Discussion section first paragraph, lines 303-304, you state that you attempted “to gather data from participants from seven different occupational titles within rehabilitation services” – need to mention these criteria upfront.

Also, there is some confusion around the dissemination of emails with the survey link. Did the facilities who agreed to participate send out emails to all employees or those in rehabilitation specifically? This is not clear. And why would researchers receive employee responses if all was anonymously done through mass emailing? Were participants told in their emails to contact you directly if they had questions or were interested in participating (i.e. lines 169-171)? How did this work regarding confidentiality? Why was consent gathered if participants who responded to the survey would do with the understanding this was by implied consent? This entire section needs clarification.

  • 6.2 Measures – part one of the survey was introduced. You should briefly introduce parts two and three which would then nicely lead into the subsections 6.2.1 and 6.2.2.
  • Suggest that 6.2.3 Procedures be re-labeled as 6.3 as procedures are different in context/content than 6. 2 on Measures.
  • Suggest 6.2.4 Analysis be re-labeled as 6.4. More detail is needed for this section on the analysis methods. Which descriptive statistics and correlational methods did you specifically use and why? What software did you use to conduct the analysis?
  • Results – over what time frame was data gathered – e.g. two/three weeks or longer? Were different approaches tried to recruit participants? Seems surprising you only got 25 participants across 36 rehab centers.
    • when using ‘most’ you also need to include the percentage in brackets, so that the reader knows exactly what you are referring to – see lines 220 and 221.
    • Section 7.1 – need to define what you mean by ‘older adults’, ‘adults’, ‘adolescents’, and ‘younger adults’. Provide age ranges or some description to be clear about your reference or understanding for these different groups of individuals.
    • Pages 8 and 9 of 21, lines 256 – 276 – you use percentage of participants as the ‘n’ value when ‘n’ really is the number of participants. You can just go with percentages but then do not use ‘n= X%’. All of these in this section of the results need to be corrected.
    • For Tables 6 through 11 there are no statements made to introduce each table. Statements are needed for the reader if only to indicate what stands out from each table’s results. Otherwise, a suggestion would be to include summary statements for each table in the results and move the tables to the appendix or as supplementary files, but do not just insert the tables and expect the reader to interpret these for you.
  • The Discussion section highlights or repeats a lot from the results. This section should also align the findings or your analysis of findings with what you found in the literature to either support your findings or otherwise, support that your findings are unique and further studies are needed to substantiate them.

You mention a few suggestions or recommendations coming out of your findings – this could be highlighted a bit more at the end of the Discussion section but indicate that further studies involving more participants would be needed to substantiate your findings and suggestions. Wonder if this paper could be considered as a pilot for the new tool and approach to exploring some critical aspects with rehab providers working with older adults with SUD? Can that angle be explored?

  • Limitations and Suggestions – because you do not have a validated questionnaire does not make it less credible. You needed to start somewhere if nothing was found in the literature – so acknowledge that your survey is unique and needs to be used in more studies to refine it and eventually validate it. I made the suggestion of referring to this study as a piloting of a new tool. This might also mitigate the limited sample size. 

You mention the limitation with sample size – this is a major limitation. You do not have the diversity you needed for this study, nor the numbers, but this is a starting point. You need to mitigate this limitation by stating that further studies are needed to use the new survey tool along with other validated tools to further explore other rehabilitation providers’ knowledge, attitudes and practices towards older adults with substance use disorders. These assessments are needed to address issues/barriers experienced by the providers and improve the care and treatment which older adults with SUD need and receive.

  • Appendix A. Survey Tool – did you have an introduction to the survey? It is important to see the introduction to the survey you used as part of the email that went out and the online survey tool as this needs to include some important considerations including ethical aspects (i.e. anonymity, confidentiality, etc.), estimated time to complete the survey, and if people can complete parts or all of survey, or do they need to complete all of the survey or at least certain parts.

Hope all these comments and suggestions are helpful.

Author Response

Comment: the entire paper needs to be edited for sentence structure, grammar, and statistical references, including the abstract

Response: Thank you for pointing this out. We agree with this comment. Therefore, we have edited the manuscript for clarity.

Comment: it seems odd that sections 2 to 5 are not under the Introduction.

Response: Thank you for pointing this out. We agree with this comment. Therefore, we have renumbered sections throughout the paper.

Comment: Need to clearly indicate if you are using this same guide or what other age range constitutes ‘older adults’ in your study (older adults as being 65 and older; American Addictions Centers, 2023)

Response: Thank you for pointing this out. We agree with this comment. Therefore, we have clarified that the same age range is used in the study. 

Comment: Within section 3 – Barriers to Treatment – mention was made in the opening lines (i.e. 77-79) that “…people abusing substances in 2022 did not receive the treatment they needed”. This seems to be aligned with the challenges most care services and patients experienced during COVID – was what you mentioned because of COVID? Please clarify the reason why people did not receive treatment needed.

Response: Thank you for pointing this out. We agree with this comment. Therefore, we have reviewed the literatrure and ensured that this is addressed later in the Barriers to Treatment section. 

Comment: As part of 6.1 Participants – you mentioned inclusion criteria for the facilities you contacted but no inclusion criteria for the participants. The latter would seem to be equally important - what criteria did you use to recruit and select participants? For example, in the Discussion section first paragraph, lines 303-304, you state that you attempted “to gather data from participants from seven different occupational titles within rehabilitation services” – need to mention these criteria upfront.

Response: Thank you for pointing this out. We agree with this comment. Therefore, we have clarified the inclusion criteria along with the recruitment procedures. 

Comment: Also, there is some confusion around the dissemination of emails with the survey link. Did the facilities who agreed to participate send out emails to all employees or those in rehabilitation specifically? This is not clear. And why would researchers receive employee responses if all was anonymously done through mass emailing? Were participants told in their emails to contact you directly if they had questions or were interested in participating (i.e. lines 169-171)? How did this work regarding confidentiality? Why was consent gathered if participants who responded to the survey would do with the understanding this was by implied consent? This entire section needs clarification.

Response: Thank you for pointing this out. We agree with this comment. Therefore, we have clarified the inclusion criteria along with the recruitment procedures. 

Comment: 6.2 Measures – part one of the survey was introduced. You should briefly introduce parts two and three which would then nicely lead into the subsections 6.2.1 and 6.2.2.

Response: Thank you for pointing this out. We agree with this comment. Therefore, we have added details about the survey.

Comment: Suggest that 6.2.3 Procedures be re-labeled as 6.3 as procedures are different in context/content than 6. 2 on Measures.

Response: Thank you for pointing this out. We agree with this comment. Therefore, we have renumbered sections throughout the paper.

Comment: Suggest 6.2.4 Analysis be re-labeled as 6.4. More detail is needed for this section on the analysis methods. Which descriptive statistics and correlational methods did you specifically use and why? What software did you use to conduct the analysis?

Response: Thank you for pointing this out. We agree with this comment. Therefore, we have added that descriptive statistics were presented in the tables as frequencies and percentages and discussed why we chose to conduct the Spearman's rank order Correlation as the most appropriate for the ordinal level data. Excel was used to analyze the data.

Comment: Results – over what time frame was data gathered – e.g. two/three weeks or longer? Were different approaches tried to recruit participants? Seems surprising you only got 25 participants across 36 rehab centers.

Response: Thank you for pointing this out. We agree with this comment. Therefore, we have clarified that data collection lasted three weeks.

Comment: when using ‘most’ you also need to include the percentage in brackets, so that the reader knows exactly what you are referring to – see lines 220 and 221.

Response: Thank you for pointing this out. We agree with this comment. Therefore, we have added the percentage in brackets as requested.

Comment: Section 7.1 – need to define what you mean by ‘older adults’, ‘adults’, ‘adolescents’, and ‘younger adults’. Provide age ranges or some description to be clear about your reference or understanding for these different groups of individuals.

Response: Thank you for pointing this out. We agree with this comment. Therefore, we have included age ranges to clarify terminology.

Comment: Pages 8 and 9 of 21, lines 256 – 276 – you use percentage of participants as the ‘n’ value when ‘n’ really is the number of participants. You can just go with percentages but then do not use ‘n= X%’. All of these in this section of the results need to be corrected.

Response: Thank you for pointing this out. We agree with this comment. Therefore, we have added the n for each percentage indicated.

Comment: For Tables 6 through 11 there are no statements made to introduce each table. Statements are needed for the reader if only to indicate what stands out from each table’s results. Otherwise, a suggestion would be to include summary statements for each table in the results and move the tables to the appendix or as supplementary files, but do not just insert the tables and expect the reader to interpret these for you.

Response: Thank you for pointing this out. We agree with this comment. Therefore, we have added references to those tables.

Comment: The Discussion section highlights or repeats a lot from the results. This section should also align the findings or your analysis of findings with what you found in the literature to either support your findings or otherwise, support that your findings are unique and further studies are needed to substantiate them.

Response: Thank you for pointing this out. We agree with this comment. Therefore, we have revised the Discussion to remove redundancy.

Comment: You mention a few suggestions or recommendations coming out of your findings – this could be highlighted a bit more at the end of the Discussion section but indicate that further studies involving more participants would be needed to substantiate your findings and suggestions. Wonder if this paper could be considered as a pilot for the new tool and approach to exploring some critical aspects with rehab providers working with older adults with SUD? Can that angle be explored?

Response: Thank you for pointing this out. We agree with this comment. Therefore, we have added a suggestion of using a larger randomly selected sample to gain a better understanding of the issue.

Comment: Limitations and Suggestions – because you do not have a validated questionnaire does not make it less credible. You needed to start somewhere if nothing was found in the literature – so acknowledge that your survey is unique and needs to be used in more studies to refine it and eventually validate it. I made the suggestion of referring to this study as a piloting of a new tool. This might also mitigate the limited sample size. 

Response: Thank you for pointing this out. We agree with this comment. Therefore, we have addressed the limitation with the sample size.

Comment: You mention the limitation with sample size – this is a major limitation. You do not have the diversity you needed for this study, nor the numbers, but this is a starting point. You need to mitigate this limitation by stating that further studies are needed to use the new survey tool along with other validated tools to further explore other rehabilitation providers’ knowledge, attitudes and practices towards older adults with substance use disorders. These assessments are needed to address issues/barriers experienced by the providers and improve the care and treatment which older adults with SUD need and receive.

Response: Thank you for pointing this out. We agree with this comment. Therefore, we have addressed the limitation with the sample size.

Comment: Appendix A. Survey Tool – did you have an introduction to the survey? It is important to see the introduction to the survey you used as part of the email that went out and the online survey tool as this needs to include some important considerations including ethical aspects (i.e. anonymity, confidentiality, etc.), estimated time to complete the survey, and if people can complete parts or all of survey, or do they need to complete all of the survey or at least certain parts.

Response: Thank you for pointing this out. We agree with this comment. Therefore, we have included the introduction to the survey that was part of the email sent out to participants. 

Reviewer 2 Report

Comments and Suggestions for Authors

The paper, titled "Understanding Rehabilitation Providers: Knowledge, Attitudes, and Practices Toward Older Adults with Substance Use Disorders," addresses an interesting topic because it addresses a frequently overlooked population with a socially relevant health problem: substance use.
The paper includes a very thorough literature review in the introduction on the topic addressed; however, it is suggested to carefully review the following aspects:
- The objective of the study is different from that stated in the abstract and at the end of the introduction.
- At the end of the introduction, it is stated: "The purpose of the current study was to examine the knowledge, attitudes, and practices of rehabilitation service providers working with older adults with substance use disorders." However: 1) an instrument on "Expectations Regarding Aging (ERA)" is included, which is inconsistent with the purpose of the study.
- On page 5, in section 6.2.4. Analysis, it states "Correlations between the ERA-12 and questions related to participants' knowledge, attitudes, and practices were conducted to explore the impact of healthcare providers' expectations regarding aging on the medical care and services they provide to older adults." However, this analysis is not consistent with the objective, and the results section reports the results of the specified analysis.
- The results presented are descriptive, so it is suggested that the objective be revised to include the ERA instrument and emphasize the relevance of the study.

Author Response

Comment: The objective of the study is different from that stated in the abstract and at the end of the introduction.

Response: Thank you for pointing this out. We agree with this comment. Therefore, we have clarified the purpose of the study as it relates to exploring the role of ageism in the provision of services to older people experiencing substance use disorders.

Comment: At the end of the introduction, it is stated: "The purpose of the current study was to examine the knowledge, attitudes, and practices of rehabilitation service providers working with older adults with substance use disorders." However: 1) an instrument on "Expectations Regarding Aging (ERA)" is included, which is inconsistent with the purpose of the study.

Response: Thank you for pointing this out. We agree with this comment. Therefore, we have clarified the purpose of the study as it relates to exploring the role of ageism in the provision of services to older people experiencing substance use disorders. As the ERA is a validated tool for measuring ageism, it is appropriate for this study.

Comment: On page 5, in section 6.2.4. Analysis, it states "Correlations between the ERA-12 and questions related to participants' knowledge, attitudes, and practices were conducted to explore the impact of healthcare providers' expectations regarding aging on the medical care and services they provide to older adults." However, this analysis is not consistent with the objective, and the results section reports the results of the specified analysis.

Response: Thank you for pointing this out. We agree with this comment. Therefore, we have revised the objective of the study.

Comment: The results presented are descriptive, so it is suggested that the objective be revised to include the ERA instrument and emphasize the relevance of the study.

Response: Thank you for pointing this out. We agree with this comment. Therefore, we have clarified the purpose of the study as it relates to exploring the role of ageism in the provision of services to older people experiencing substance use disorders. As the ERA is a validated tool for measuring ageism, it is appropriate for this study.

Reviewer 3 Report

Comments and Suggestions for Authors

See attached file

Comments on the Quality of English Language

Consider professional editing services

Author Response

Comment: The introduction (1 to 5) is too long.

Response: Thank you for pointing this out. We agree with this comment. Therefore, the introduction (section 1) has been edited for clarity and brevity when possible while still providing the relevant background information. 

Comment: Row 84 rewrite Boekel and researchers

Response: Thank you for pointing this out. We agree with this comment. Therefore, we have revised the paper to say "Boekel and colleagues".

Comment: Avoid repetition. rows 110-119 Odani

Response: Thank you for pointing this out. We agree with this comment. Therefore, we have revised our discussion of Odani et al.'s work in section 1.3 to reduce repetition.

Comment: Unclear why "This study was determined to be exempt from the <blinded for review> Institutional Review Board"

Response: Thank you for pointing this out. We clarified that this study was exempt from IRB review because it was completely anonymous and posed no/minimal risk to participants.

Comment: Very small sample. state the study design.

Response: Thank you for pointing this out. We agree with this comment. Therefore, we have stated the study design.

Comment: Was the survey pretested/validated?

Response: Thank you for pointing this out. We agree with this comment. Therefore, we have indicated that the survey was vetted by a committee of three faculty in gerontology and public health but was not a validated instrument.

Comment: The majority of respondents (n= 52.0%) row 256 – 52% is hardly a majority… use n or percentage , also in the following sentence

Response: Thank you for pointing this out. We agree with this comment. Therefore, we have revised our use of the word "majority" throughout the Results section.

Comment: Adequate modelling method. The detailed model description enables study replication.

Response: Thank you for pointing this out!  It is nice to receive positive feedback!

Comment: Results & Discussion (n= XX.0%) in all this section

Response: Thank you for pointing this out. We agree with this comment. Therefore, we have updated the results and discusison sections with the count and percentage as appropriate.

Comment: Indicate n for each table since is not 25/24 in all of them

Response: Thank you for pointing this out. We agree with this comment. Therefore, we have have added n for each table. 

Comment: Table 3 Consider rephrasing I feel confident in my knowledge of older adults. Unclear

Response: Thank you for pointing this out. We agree with this comment. Therefore, we have clarified that that is the language from the survey. 

Comment: Clarify Other in table 5

Response: Thank you for pointing this out. We agree with this comment. Therefore, we have clarified that "Other" was included to recognize that there are strategies and interventions used by providers that were not listed as response options. 

Comment: Row 290 0 explain the score of 50

Response: Thank you for pointing this out. We agree with this comment. Therefore, we have clarified the score (n=12;48.0%). 

Comment: Conclusions section contains ideas already present in Discussions section.

Response: Thank you for pointing this out. We agree with this comment. Therefore, we have rearranged content in the discussion and conclusion.

Comment: Limitations and Suggestions - Lacks mentioning bias

Response: Thank you for pointing this out. We agree with this comment. Therefore, we have addressed the response bias. 

Comment: Include cut-offs’ for ERA12

Response: Thank you for pointing this out. There are no established cutoff scores for the ERA-12. We included this information in section 2.2.2.

Comment: N is not %

Response: Thank you for pointing this out. We agree with this comment. Therefore, we have revised this thoughout to indicate the correct count or percentage as appropriate.

Comment: Only descriptive statistics. Lack at least Correlations.

Response: Thank you for pointing this out. We agree with these comments. Therefore, we addressed this by conducting Spearman's correlations and indicating that none were statistically significant.

Round 2

Reviewer 1 Report

Comments and Suggestions for Authors

Thank you for the revisions and edits made. They certainly improved the manuscript.

Author Response

Comment: Thank you for the revisions and edits made. They certainly improved the manuscript.

Response: We appreciate the thorough review and agree the manuscript is much improved. Thank you for your kind words here.

Reviewer 3 Report

Comments and Suggestions for Authors

most of the issues have been addressed.  Limitations and Suggestions still lacks mentioning bias linked to self-administration . no changes here although the authors stated that changes have been made

Author Response

Comment: most of the issues have been addressed.  

Response: Thank you.  We worked hard to address the extensive feedback provided and feel the manuscript is much improved.

Comment: Limitations and Suggestions still lacks mentioning bias linked to self-administration . no changes here although the authors stated that changes have been made.

Response: We originally added mention of self-selection bias (people chose to participate because they had an interest in the topic) but realize that isn't what you were asking for. We have now included description of bias related to self-report (decision style) with a suggestion to address that issue in future studies with a larger sample.